# Findings and Prognosis in 149 Horses with Histological Changes Compatible with Inflammatory Bowel Disease

**DOI:** 10.3390/ani14111638

**Published:** 2024-05-30

**Authors:** Lieuwke Cecilia Kranenburg, Bo F. Bouwmeester, Robin van den Boom

**Affiliations:** Department of Clinical Sciences, Faculty of Veterinary Medicine, Utrecht University, Yalelaan 112, 3584 CM Utrecht, The Netherlands; l.c.kranenburg@uu.nl (L.C.K.);

**Keywords:** equine, horse, inflammatory bowel disease (IBD), enteritis

## Abstract

**Simple Summary:**

Inflammatory bowel disease (IBD) is a chronic intestinal disease in horses, causing weight loss, reduced performance and recurrent (mild) colic. Different types of inflammatory cells infiltrate the intestine and characterize several different forms of IBD. Biopsies were obtained from the first portion of the small intestine during endoscopy of the stomach and studied microscopically. When increased numbers of inflammatory cells were present, the horses were included in the present study. The microscopic changes were classified as mild, moderate or severe and the predominant infiltrating cell type was recorded. Clinical improvement was assessed by the owners at 6 weeks after biopsy, along with survival after one year. In total, 149 horses were included, and the most common clinical signs were weight loss, reduced performance and pain during abdominal palpation. The uptake of glucose from the small intestine was impaired in most horses with IBD, and the horses with severe IBD had lower protein concentrations in their blood. Overall, 71% of the cases had improved clinically after six weeks, mostly following treatment with corticosteroids. The results of a second biopsy were a poor predictor of improvement and the horses that improved after 6 weeks were more likely to be alive after one year.

**Abstract:**

Inflammatory bowel disease (IBD) is a chronic disease characterized by different cell infiltrates in the intestine. The aims of this study were to report the clinical and clinicopathological findings in horses with histological changes compatible with IBD in the duodenum. Further, the clinical progression of IBD and survival were investigated. Patient records were reviewed for horses in which histological evidence of IBD was found in duodenal biopsies collected during endoscopy. The histological changes were classified as mild, moderate or severe and the predominant infiltrating cell type was recorded. Clinical improvement was assessed by the owner via a questionnaire at 6 weeks after biopsy, along with survival after one year. In total, 149 horses were included, and the most common clinical signs were weight loss, reduced performance and pain during abdominal palpation. Most horses showed partial malabsorption during an oral glucose absorption test, and the horses with severe IBD had lower serum protein concentrations. Lymphoplasmacytic enteritis was the most common type of IBD (78.5% of cases), while in six horses neutrophilic infiltration of the duodenum was present. Overall, 71% of the cases had improved clinically after six weeks, mostly following treatment with corticosteroids. The results of a second biopsy were a poor predictor of improvement, and the horses that improved after 6 weeks were more likely to be alive after one year.

## 1. Introduction

Inflammatory bowel disease (IBD) in horses is an umbrella term for all intestinal diseases with infiltration of different cell types of the intestines and with similar clinical signs [1,2,3]. Most commonly, the small intestine is affected, although the large intestine can also be involved [1,2,3]. IBD is a common cause of weight loss despite a good appetite in horses. In one study, the prevalence of IBD in horses with weight loss was 32.5% [4]. The weight loss in horses with IBD is mainly caused by malabsorption and maldigestion [1,2,3,5]. Other symptoms are ventral edema, lethargy, diarrhea and abdominal pain, and occasionally ventral edema [1,5]. Although the etiology is incompletely understood, there are some indications that immune-mediated processes or infectious causes may play a role [1,2]. An association was found between IBD in horses and elevated concentrations of gluten-dependent antibodies [6].

Depending on the infiltrating cell types, IBD can be divided into four different types—granulomatous enteritis (GE), lymphocytic–plasmacytic enterocolitis (LPE), multisystemic eosinophilic epitheliotropic disease (MEED) and idiopathic eosinophilic enterocolitis (EC) [1,2,3]—with some variation in symptoms. Distinguishing between the four different types is only possible with a histopathological examination of the intestine [1,2]. 

Granulomatous enteritis (GE) can be compared to Crohn’s disease in humans [7]. It mainly causes weight loss, anorexia, anemia and hypoalbuminemia [1,2]. The anemia can be immune-mediated or related to the malabsorption of elements needed for erythropoiesis [1,8]. The main cause of hypoalbuminemia is an enteric loss of protein [1,9,10]. Standardbreds and young horses seem to be predisposed to GE, since about 80% of the horses with GE are Standardbreds and almost 90% are aged less than five years [1,2,3,9]. 

In jejunal full-thickness biopsies, rectal biopsies and histopathological post-mortem findings of the ileum sheets of macrophages, lymphoid infiltrations and other epithelioid cells are found in the lamina propria and submucosa [1,3,7,11]. Many plasma cells and giant cells are found in the ileum post-mortem in some cases of horses with GE [3,7,11]. Circumscribed granulomas in the (sub)mucosa, without other inflammatory cells or etiological agents, are typical pathological findings in rectal biopsies [1,2]. Besides these cellular infiltrates, villous atrophy can be found in horses with GE [1,8]. In some cases, horses with GE also have gastric lymphocytic, histiocytic infiltration or ulcers [8,10]. 

The least commonly reported type of IBD in horses used to be lymphocytic–plasmacytic enterocolitis (LPE) [1]. In the more recent literature, however, it appears that LPE is the most common form of IBD [5,12]. LPE causes weight loss, variable diarrhea, depression, recurrent colic and inappetence [2,13,14]. The mucosal infiltration of lymphocytes and plasma cells, without granulomatous changes and villous atrophy, can be found using a histopathological examination of the affected intestine [2,14]. However, these cells can also be found with many other intestinal diseases [13]. 

Multisystemic eosinophilic epitheliotropic disease (MEED) causes dermatological, pancreatic, hepatic and gastrointestinal lesions [1,2,3]. The oral cavity, lungs and mesenteric lymph nodes can also be affected, and the skin lesions are mainly located on the head, limbs, ventrum and coronets [3]. These additional findings in horses with MEED can help distinguish them from horses with GE, although some horses with GE also have skin lesions [1,2]. As with GE, Standardbreds and young horses (four years or younger) seem to be predisposed to developing MEED [1,2,3]. Elevated concentrations of alkaline phosphatase and gamma glutamyl transferase (GGT), indicative of damage to the liver, are often found in horses with MEED, and this can help distinguish MEED from GE [1,2,3]. Horses with MEED are occasionally anemic, in contrast to horses with GE, which are usually anemic [1]. Eosinophilic, lymphocytic and sometimes basophilic infiltrations can be found in the mucosa of enteral biopsies of horses with MEED, although villous atrophy is rarely seen [2,3,9].

Another form of IBD is eosinophilic enterocolitis (EC). This type should be classified separately, as it causes other clinical signs than the other forms of IBD and it has a better prognosis [1]. EC mainly causes (recurrent) colic, without weight loss [1,2]. Colic is caused by obstruction of the lumen of the small or (rarely) large intestine through circumferential mural bands due to eosinophilic enzymes that stimulate mural fibrosis, as is also described in horses with MEED [2,3,15]. At the time of post-mortem histopathological examination, eosinophilic, lymphocytic or eosinophilic granulomatous infiltrates with fibrinoid necrosis of intramural vessels can be found in the lamina propria and submucosa of the intestine [2,3]. 

In addition to the history and clinical signs, thickened loops of small intestine in the rectal palpation or transabdominal ultrasound results can help establish a diagnosis of IBD [3,16]. A final diagnosis is hard to obtain because no conclusive diagnostic test is currently available [5,16]. The minimally invasive methods to obtain more information about the intestinal function are the oral glucose absorption test (OGAT) and D-xylose test, which can be used to identify partial or total malabsorption [3,17]. The D-xylose test is a more specific test and is not affected by metabolic or hormonal influences [3,18]. The D-xylose test seems to distinguish horses with GE from horses with EC, since in a study all nine horses with GE had a significantly lower mean D-xylose absorption value than the controls and all nine horses with EC did not show a significant difference in mean D-xylose absorption compared to the controls [9]. Kaikkonen et al. found that the peak xylose concentration in horses with IBD was higher in survivors than non-survivors [19].

Horses with IBD often have hypoproteinemia, due to the decreased absorptive function of their (small) intestines due to the cell infiltrates [2,9,10], and at least 70% of the protein digestion in horses is prececal [20]. 

Only histology of enteral biopsies can provide a definitive morphological diagnosis [1,5], while biopsies of the skin and liver can be helpful in diagnosing MEED [2,3]. However, the interpretation of biopsies by pathologists is subjective and the findings can be interpreted differently [21]. Enteral biopsies can be obtained in three ways—from the rectum, via gastro(endo)scopy (of the duodenum) and during surgery (laparotomy and laparoscopy). Rectal biopsies have been shown to be useful in diagnosing intestinal disorders, such as IBD, in horses, since pathological changes were found in 52% of horses with intestinal diseases. Rectal biopsies are considered to provide an appropriate diagnosis in approximately half of the horses with GE or MEED but are not useful in confirming LPE or EC [1, 22.]. Such biopsies should be examined by an experienced pathologist because inflammatory cells are often found in the rectal tissue of clinically normal horses [1,22]. 

Small intestinal samples are preferred over large intestinal samples [21], and these can be obtained via endoscopy, although only from the proximal duodenum [3]. When the lesions are present further along in the duodenum or in other parts of the small intestine, they can be missed [3]. Another disadvantage of duodenal biopsies obtained by endoscopy is that they are small and not full-thickness, and their diagnostic value is currently unclear [1,5,22]. 

The greatest degree of certainty regarding a diagnosis of IBD is obtained via a histological examination of a full-thickness intestinal biopsy, obtained at either the time of surgery or post-mortem examination [3,13]. 

The treatments for IBD can be divided into dietary changes, medical therapy and surgery [1,2,3]. The prognosis of IBD in horses is generally considered to be poor [1,2,3,23], although in a more recent study a 65% three-year survival rate was found [19]. The survival rate was significantly higher in horses that responded to anthelmintic and corticosteroid therapies within three weeks than in horses that did not respond to the initial treatment [19]. 

The aim of the dietary changes is increased digestibility and sometimes to eliminate possible irritating antigens [3]. Evidence for a role of diet in IBD was provided by a recent study in which horses on a low-fiber diet developed greater cellular infiltration in the intestine than those on a high-fiber diet [24]. Greater digestibility can be achieved via more frequent feeding, as well as the consumption of smaller portions and special diets composed of hay, oats and corn oil [3]. A mono diet can help in identifying and removing possible antigens [3]. When a gluten allergy is proven, a gluten-free diet and management strategies can reduce the clinical signs [6]. 

The aim of medical therapy is to reduce inflammation and eliminate antigens, and it can consist of corticosteroids, antibiotics, anthelmintics or a combination of these [1,3]. Most horses suffering from IBD are treated with corticosteroids. The long-term prognosis for horses with IBD treated with corticosteroids is reported to be poor, although full clinical remission has been reported [1,19,23,25]. Eosinophilic and lymphocytic or plasmacytic cases may respond better to corticosteroid therapy than granulomatous cases [2]. Antimicrobial and anthelmintic drugs are also sometimes used to treat IBD in horses, as infectious agents may play a role in the etiology of IBD, although their efficacy is unclear [1,2,3]. 

Another alternative drug in the treatment of horses with IBD is azathioprine. It is commonly used in people with IBD and has proven to be immunosuppressive in horses with immune-mediated diseases [26,27,28]. However, only a single case report describes the successful treatment of horses with IBD with azathioprine [29].

The surgical removal of affected pieces of intestine is associated with variable results [1,2]. In a study with two horses suffering from GE that underwent intestinal resection, one was euthanized because of recurring colic and the other was clinically normal after 13 months [30]. The surgical removal of eosinophilic mural bands and other local eosinophilic lesions in horses with MEED or EC has proven to be effective, as in one study five out of six horses were alive 5–60 months after surgery [31]. In another study, seven out of ten horses recovered completely from EC after surgery [32]. However, horses with EC can recover with only corticosteroid therapy in many cases [1,33,34].

We have the impression that many horses that are diagnosed with IBD also suffer from equine gastric ulcer syndrome (EGUS), especially equine glandular gastric disease (EGGD). Boshuizen et al. investigated a possible association between IBD and EGUS but did not find one [5].

Compared to humans and companion animals, far less scientific evidence is available for IBD in horses. The lack of insight into the etiology and pathophysiology of IBD complicates the interpretation of the diagnostic findings and the implementation of rational treatments. In addition, the prognoses of equine IBD reported in the literature vary considerably, making it difficult to advise owners. We hypothesized that the prognosis of IBD in horses is better in animals with less severe clinical signs, higher serum total protein rates, a normal oral glucose absorption test result, less severe histological changes in a duodenal biopsy, an improved histology score in a second biopsy and receiving treatment with corticosteroids. In addition, the type of IBD may also influence the prognosis.The aim of this study was to evaluate whether associations could be found between the clinical signs, histological biopsy results, treatment and prognosis of IBD in horses. As an association between IBD and equine gastric ulcer syndrome (EGUS) is often suspected, an additional aim was to explore a possible association between the presence and type of IBD and the gastroscopic findings.

## 2. Materials and Methods

### 2.1. Study Population

Patient records of horses that underwent a duodenal biopsy at the Equine Hospital of the Faculty of Veterinary Medicine, Utrecht University, the Netherlands between June 2010 and June 2020 were reviewed. The inclusion criteria were horses diagnosed with IBD based on the histological evaluation of the duodenal biopsy. Horses suspected of IBD but with no duodenal biopsy were excluded from the study. The age at diagnosis of IBD, sex, breed and clinical signs were recorded. The clinical signs were divided into ‘weight loss’, ‘diarrhea’, ‘colic (once)’, ‘colic (recurrent)’, ‘decreased appetite’, ‘reduced performance’, ‘edema’, ‘sensitive abdomen’ and ‘other’.

An informed consent form was signed by the owner at the time of admission to the clinic, allowing the use of the data for scientific purposes.

### 2.2. Blood Work

The total serum protein concentration was recorded. When an oral glucose absorption test was performed, using 1 g/kg bodyweight glucose administered by nasogastric tube, the maximal percentage increase in glucose concentration in the blood was noted. Two cut-off values were used to indicate (partial) malabsorption—a blood glucose concentration increase of less than 85% or less than 50%.

### 2.3. Duodenal Biopsies and Endoscopic Findings

The duodenal biopsies were all harvested at the Equine Hospital of the Faculty of Veterinary Medicine from sedated horses, using biopsy forceps inserted through an endoscope. The biopsy samples were stored in formalin until examined by a veterinary pathologist. All infiltrating cell types (macrophages, lymphocytes, plasma cells, eosinophils and neutrophils) were recorded, and on the basis of the predominant infiltrating cell type, the type of IBD was classified as ‘LPE’, ‘EC’, ‘MEED’ or ‘a combination of LPE and EC’. Additionally, the last three categories were combined into one group named ‘eosinophilic’. The severity of the IBD was rated by the pathologists based on histological evaluation of the biopsies and categorized as ‘mild’, ‘moderate’ or ‘severe’. In approximately 20% of cases, a second biopsy was obtained to investigate the effect of the treatment on duodenal inflammation. The decision on whether or not to take a second biopsy was made together with the owner, and it was collected approximately 6 weeks after the initial biopsy and the start of therapy and examined as described above.

During gastroscopy, which was performed when taking the duodenal biopsies, the presence and severity of equine squamous gastric disease (ESGD) and equine glandular gastric disease (EGGD) were recorded. It was also noted whether the horse received medication (omeprazole or sucralfate) for the treatment or prevention of gastric ulcers after taking the biopsies.

### 2.4. Treatment

The initial treatment of horses with IBD at the Equine Hospital of the Faculty of Veterinary Medicine, Utrecht University, usually consists of four to six weeks of corticosteroids, either prednisolone (per os) or dexamethasone (intramuscularly) followed by prednisolone (per os). Sometimes, dexamethasone is only given as an initial treatment, so the medical therapies were divided into ‘prednisolone’, ‘dexamethasone and prednisolone’, ‘dexamethasone’ and ‘none’. In addition to the medical therapies, any dietary changes that were advised were noted. The dietary changes were divided into ‘gluten-free diet’, ‘other’ and ‘none’. The dietary changes described as ‘other’ often consisted of an increase in the amount of forage or a reduction in the amount of compound complementary feed.

### 2.5. Outcome

The outcome was assessed six weeks after diagnosis and the initiation of treatment of IBD based on an assessment by the owner of the severity of clinical signs. The outcomes were divided into ‘improved’ and ‘not improved’. A patient was considered ‘improved’ when no clinical signs related to IBD were present or when they displayed fewer or less severe clinical signs compared to before the diagnosis of IBD was made. The group ‘not improved’ included horses with the same severity of clinical signs, more severe signs or when the clinical signs had improved but then recurred within six weeks.

The survival result was recorded a year after diagnosis. If a horse had died, the cause of death was recorded as (a) euthanized for reasons related to IBD, (b) euthanized for reasons not related to IBD or (c) natural death.

When a second biopsy was obtained, the severity was assessed by the veterinary pathologist and then compared to the initial biopsy result. The second biopsy was also categorized as ‘improved’ or ‘not improved’ based on the histological examination.

### 2.6. Statistical Analysis

A one-way-ANOVA was used to determine whether the increase in blood glucose concentrations and the total serum protein concentrations differed significantly between horses with a different severity of IBD, horses with different types of IBD and horses that had and had not improved six weeks after diagnosis. The effects of different clinical signs, age, the total serum protein concentration, the percentage increase in glucose in an OGAT, histological severity, the presence of ESGD, the presence of EGGD, the type of IBD and the medical treatment on the clinical improvement at six weeks was investigated using logistic regression. Akaike’s information criterium (AIC) was used for model reduction. For the effects in the final model, the profile confidence intervals were calculated. A Chi square test was performed to determine whether the outcomes after one year differed between the horses that had improved after six weeks and the horses that had not improved after six weeks.

## 3. Results

### 3.1. Study Population Characteristics

In total, 149 horses diagnosed with IBD by duodenal biopsy were included in the study. Of these 149 horses, 84 (56.4%) were geldings, 61 (40.9%) were mares and four (2.7%) were stallions.

The study population consisted of 100 Warmblood horses (67.1%), nineteen ponies and coldblooded horses (12.8%), ten Friesian horses (6.7%) and nineteen horses of various other breeds (12.8%), while in one case the breed was not recorded. The minimum age at admission was two years and the maximum age was twenty-two years, with a median age of ten years.

#### Clinical Signs

The percentages of horses demonstrating various clinical symptoms for each type of IBD are shown in Figure 1. The most common symptoms were weight loss, reduced performance and a sensitive abdomen.

### 3.2. Blood Work

Fifty-four horses underwent an oral glucose absorption test, and the results are shown in Figure 2. The increases in glucose concentration in the blood varied from 18% to 100%, and different cut-off values for (partial) malabsorption are reported in the literature. If a normal increase in glucose concentration in response to an oral glucose challenge is considered to be ≥50%, 18/54 (33%) horses with IBD were classified as having (partial) malabsorption. In 22 horses (41%) the glucose concentration increased by 50% to 85%. If a normal increase in glucose concentration is defined as ≥85%, 40/54 (74%) horses with IBD were classified as having (partial) malabsorption.

The increases in glucose concentration in response to the OGAT for different severity groups and different types of IBD and the outcomes (improved or not improved) after six weeks are shown in Figure 2. No association was found between an increase in glucose concentration and the severity of IBD (F = 0.29; *p* = 0.746) or the outcome after six weeks (F = 0.80; *p* = 0.376). An association was found between an increase in glucose concentration and the type of IBD (F = 3.53; *p* = 0.021). Horses with LPE (*p* = 0.027) and neutrophilic enteritis (*p* = 0.046) showed a smaller increase in glucose concentration than the horses with EC. No significant difference was found in the increases in glucose concentration between the horses with EC and the combined form of LPE and EC (*p* = 0.813).

The total serum protein concentrations were measured in 113 horses, and the mean concentration was 62.4 g/L (minimum 33 g/L, maximum 78 g/L). If hypoproteinemia is defined as <52 g/L, 12 horses (11%) were classified as hypoproteinemic. The total serum protein concentrations in horses with different severity of IBD, types of IBD and outcomes are shown in Figure 3. A significant difference (F = 3.77; *p* = 0.026) was found in the total serum protein concentrations in the different severity groups. Horses with severe IBD had a significantly lower total serum protein concentration (*p* = 0.011) than horses with mild IBD. No significant difference (*p* = 0.113) in total serum protein concentrations was found between horses with mild and moderate IBD. No association was found between the total serum protein concentrations and the different types of IBD (F = 1.71; *p* = 0.152) or the outcomes after six weeks (F = 0.80; *p* = 0.373).

### 3.3. Duodenal Biopsies and Gastroscopic Findings

Based on the biopsies 117 horses (79%) suffered from LPE, 18 horses (12%) had a combined form of LPE and EC, 7 horses (5%) had EC, 1 horse (1%) suffered from MEED and 6 horses (4%) had neutrophilic enteritis. No cases of granulomatous enteritis (GE) were identified in the present study and no other combinations besides LPE and EC were observed. Seventy horses (47%) had a mild form of IBD, 66 horses (44%) had a moderate form and 13 horses (9%) had a severe form according to the pathologists. These data are summarized in Table 1.

Equine squamous gastric disease (ESGD) was found in 78/149 (52%) horses and equine glandular gastric disease (EGGD) in 89/146 (61%) horses. More detailed data about the presence of ESGD and EGGD in horses with different types of IBD and in different severity groups are presented in Table 2.

### 3.4. Treatment

Ninety-five horses (64%) were treated with prednisolone only, 12 horses (8%) were treated with dexamethasone followed by prednisolone, 6 horses (4%) were treated with dexamethasone only, 34 horses (23%) were not treated and in 2 cases (1%) the medical therapy was unknown. The horses that were not treated had mild clinical signs that did not warrant treatment with corticosteroids or the owners chose not to treat them because of a concern regarding the possible development of laminitis with steroid treatment. The percentages of the various treatments were very similar for the different severity groups, as can be seen in Table 3.

### 3.5. Outcome after 6 Weeks

The clinical outcome at six weeks was determined for 140 horses. Of these, 100 horses (71%) had improved clinically after six weeks and 40 horses (29%) had not improved. No association was found between improvement after six weeks and age. The presence (initially) of different clinical signs was not related to the improvement after six weeks.

When the cut-off for malabsorption was set as an increase in glucose concentration of less than 50% in an OGAT, 13/18 (72%) of the horses with malabsorption had improved after six weeks compared to 23/34 (68%) of the horses without malabsorption. When the cut-off value for malabsorption was set as an increase in glucose concentration of less than 85% in an OGAT, an improvement after six weeks was seen in 27/38 (71%) of the horses with malabsorption and in 9/14 (64%) of the horses without malabsorption. Of the horses with hypoproteinemia, 8/11 (73%) had improved after six weeks compared to 65/97 (67%) of those with a normal total serum protein concentration. No association was found between the improvement after six weeks and the result of an OGAT or the total serum protein concentration.

Fifty-four/73 (74%) horses with ESGD had improved after six weeks versus 46/67 (69%) horses without ESGD. For horses with EGGD, 55/82 (67%) had improved after six weeks, while 43/55 (78%) of those without EGGD had improved. There was no significant association between a clinical improvement after six weeks and the presence of gastric ulcers (ESGD or EGGD).

Clinical improvements were observed in 75%, 72% and 50% of cases of mild, moderate and severe IBD, respectively. Although the likelihood of improvement appears smaller for horses with severe IBD, the number of cases was small and the difference was not statistically significant. Similar rates of clinical improvement of 75%, 76% and 43%, were recorded for mild, moderate and severe LPE, respectively. Again, these differences were not statistically significant.

An improvement after six weeks was seen in 81/111 (73%) horses with LPE, 12/18 (67%) horses with a combined form of LPE and EC, 3/5 (60%) horses with EC and 3/5 (60%) horses with neutrophilic enteritis and in the horse with MEED. No association was found between the improvement after six weeks and the type of IBD.

The proportions of horses that had improved clinically after six weeks following treatments with prednisolone only, dexamethasone followed by prednisolone and dexamethasone only were 70%, 73% and 50%, respectively. Interestingly, the proportion of horses that received no treatment but improved was 77%. No association was found between an improvement after six weeks and the different medical therapies.

### 3.6. Outcome after One Year

The survival result after one year could be ascertained for 62 cases. Thirty-two of these horses (52%) were alive after one year, 20 horses (32%) had been euthanized for reasons related to IBD and 10 horses (16%) had died naturally or been euthanized for reasons unrelated to IBD. The likelihood of survival at one year was significantly greater for horses that had improved after six weeks than for those that had not improved clinically at 6 weeks (χ^2^ = 20.248; *p* < 0.001).

### 3.7. Second Biopsies

A second biopsy was taken in 31 cases. The histological severity of the second biopsy was less than of the initial biopsy in 13 horses (42%) but was the same or more severe in 18 horses (58%). Of the 13 horses with a less severe second biopsy, 10 (77%) had improved clinically after six weeks compared to 12/18 (67%) of horses in which the severity of the second biopsy had not improved.

## 4. Discussion

For this study, data from 149 horses of different breeds and ages with IBD were compared. However, being a retrospective study, the main limitation was the inhomogeneity of the data that were available. Fewer data were available for most variables because not all diagnostics had been performed in all cases. Most of the variables could be determined objectively and could be measured at follow-up in the same way. The only more subjective, or semi-quantitative, parameters were the severity of the biopsies, clinical signs and clinical improvement.

No differences were seen in sex, breed or age when comparing the different types of IBD. The only finding that was striking was the fact that roughly 75% of the horses with LPE were Warmbloods, while the majority of animals with a combined form of LPE and EC were ponies or coldblooded horses. Of the horses with only EC, the majority were Warmbloods, although a large proportion also consisted of Friesian horses. These differences could be explained by the small number of horses with combined LPE and EC and only EC and the fact that Warmbloods are the most common breed in our clinic.

The breeds included in the present study reflect the general hospital population. This may have affected the types of IBD found, as some breeds are considered to be predisposed to the development of certain types of IBD. For example, very few Standardbreds are seen in our hospital, and this may explain why we did not encounter any cases of GE. However, there were more Warmbloods in the current study population than in the general hospital population (67% vs. 50%), suggesting that Warmbloods may be more likely than other breeds to develop IBD. Additionally, ours is a referral hospital, meaning our findings may not be (directly) extrapolated to the general horse population.

In the literature, only predispositions for GE and MEED have been reported, both in Standardbreds and young horses [1,2,3,9]. In this study, no horse with GE was diagnosed and only one horse with MEED was diagnosed. The horse with MEED was relatively young (5 years old), although it was not a Standardbred.

The most common clinical sign in horses with IBD was weight loss, although reduced performance and a sensitive abdomen were also common signs (Figure 1). Recurrent colic and decreased appetite were seen in almost half of the cases. Diarrhea was seen in approximately one-third of the horses with IBD and a single episode of colic and edema was seen in only about 10% of horses. Weight loss is also the most common sign according to the literature [1,2,3,4,5]. Schumacher et al. reported abdominal pain as the most common sign associated with EC, while no mention of abdominal pain was made in horses with LPE [1]. In the present study, the percentages of horses demonstrating a sensitive abdomen were almost the same for the different types of IBD. Lethargy or depression are also commonly reported clinical signs in horses with IBD, especially those with LPE [1,2,13,14]. However, in the study by Boshuizen et al., only 22% of horses were considered to be lethargic [5].

For the OGAT, two different cut-off values were used to indicate the presence of (partial) malabsorption, since there is no consensus in the literature regarding the cut-off value that is most reliable [35]. If an increase in glucose concentration of 85% or more is considered normal, as described by Boshuizen et al. and Mair et al. [5,17], (partial) malabsorption was found in 74% of the horses that underwent an OGAT. This result is very similar to that reported by Boshuizen et al., who found decreased absorption following an oral glucose tolerance test (OGTT) in 70.5% of horses [5]. In the current study, the smallest increase in glucose concentration was 18%. If the limit for total malabsorption is set at 15%, as in the study by Mair et al., no horse suffered total malabsorption [17]. To distinguish between mild and more severe malabsorption, we also looked at horses with an increase in glucose concentration of less than 50%. When that cut-off value was applied (partial), malabsorption was seen in only 33.3% of horses. In the present study, similar values for the minimum, mean and maximum glucose concentration increases were found in cases of mild and moderate IBD, while the mean increase in glucose was approximately 10% lower in horses with severe IBD. However, this difference was not statistically significant. In addition, horses with EC or a combined form of EC and LPE showed a significantly larger increase in glucose concentration than horses with LPE or neutrophilic enteritis. These data suggest that eosinophilic enteritis has less impact on intestinal absorption than LPE and neutrophilic enteritis, possibly because the latter forms affect a larger proportion of the small intestine. The OGAT may be affected by other conditions than just IBD, including endocrinological diseases such as equine metabolic syndrome (EMS). This seems unlikely in the horses included in the present study, although a xylose absorption test may be more specific for the identification of malabsorption in horses. However, our laboratory does not offer plasma xylose measurements.

A small but significant difference was seen in the mean and maximum serum protein concentrations, with those in severe cases being lower than in cases of mild IBD. This finding indicates that the serum protein concentration may be a measure of IBD severity. However, the mean serum protein concentration was not below 52 g/L, the value regarded as indicating hypoproteinemia, in any of the severity categories.

The severe form of IBD was diagnosed four times as often in horses with a combined form of EC and LPE than in horses with just LPE. No horse with EC suffered severe IBD. These differences could be explained by the small number of horses with the severe form or the fact that having two different types of IBD concurrently indicates more extensive inflammation. Overall, horses with severe IBD demonstrated a greater impairment of their glucose absorption, while those with severe combined IBD (LPE and EC) showed a larger increase in glucose concentration after the OGAT. This could again be explained by the fact that even severe EC affects a smaller proportion of the small intestine.

Of the 149 horses in the present study, 117 (78.5%) suffered from LPE. This contradicts the earlier literature, which suggested that LPE is the least common form of IBD in horses [1,14]. Until 2017, only 24 cases of LPE had been reported [13,14,19,23,36,37,38]. A more recent study found another 16 horses with LPE out of 37 horses with IBD in which a duodenal biopsy was taken [5]. That same study also found 36 horses with LPE out of 45 horses with IBD in which a rectal biopsy was taken [5]. However, in some cases both duodenal and rectal biopsies were taken, so the exact total number of horses with LPE in that study cannot be calculated [5]. LPE is often diagnosed in dogs and cats with IBD [39,40,41,42], and canine LPE is suggested to be a nonspecific intestinal immune response to different agents that cause intestinal damage [39]. Given that many of the body’s response mechanisms are similar in multiple (mammal) species, the same could apply to horses.

Six horses diagnosed with IBD in the present study showed only neutrophilic enteritis in their duodenal biopsies. These horses were regarded and treated as horses with IBD, although no cases of IBD in horses involving only neutrophilic infiltrates have previously been described. Neutrophilic infiltration without other cell infiltrates is seen in a subset of cats with IBD, and *Campylobacter coli* is associated with this form of IBD [41]. Neutrophilic infiltrates can also be found in combination with other cell infiltrates in dogs with IBD [43]. In horses, intestinal neutrophilic infiltrates can be found in cases of duodenitis–proximal jejunitis (DPJ) [44,45]. Duodenitis–proximal jejunitis is an acute disease that mainly manifests as colic, gastric reflux and decreased intestinal motility [44,45]. Four of the six horses with neutrophilic enteritis in the present study showed recurrent colic, rather than a single bout of colic, and none had gastric reflux. All of the horses with neutrophilic enteritis suffered weight loss, suggesting a more chronic intestinal disease like IBD. The etiology of DPJ is not completely understood, although it is associated with the presence of *Salmonella*, mycotoxins and *Clostridium* species [44,45]. Considering these examples of DPJ in horses and IBD with neutrophilic infiltration in cats associated with *Campylobacter coli*, it could be that IBD in horses with neutrophilic infiltration is associated with bacteria.

As IBD is considered to be an inflammatory condition, which is possibly immune-mediated, it is usually treated with corticosteroids. Most of the horses in the present study population received prednisolone, sometimes preceded by dexamethasone. A small number of horses were only treated with dexamethasone. Dexamethasone is generally considered to be more potent than prednisolone, and it might have been expected that horses with more severe IBD would be treated more often with dexamethasone. The route of administration, being oral for prednisolone and intramuscular for dexamethasone, may have also influenced the choice for one steroid or another. A limitation of this study is the large variation in the number of horses receiving different treatments, making comparisons of these treatment options very difficult.

The overall prognosis of IBD seemed to be better than previously reported in the literature, since 100 (71.4%) horses had improved clinically after six weeks and 51.6% of the horses available for follow-up were still alive one year after diagnosis. It is possible that the one-year survival data were negatively affected by the fact that horses with recurring clinical signs are more likely to return to the clinic and be available for follow-up. The specific clinical symptoms demonstrated did not influence the outcome after six weeks. This could be because the severity of the symptoms was not taken into account. The development of an equine IBD severity score would be useful in assessing responses to treatment in the future. Interestingly, the horses that received no medical therapy were just as likely to have improved after six weeks than those treated with corticosteroids. This could not be explained by the fact that medical treatment was withheld in horses with mild IBD, as that was not the case. The presence of gastric ulceration or the administration of omeprazole or sucralfate did not affect the outcome at six weeks. This may be seen as evidence that the clinical signs were related to IBD, rather than gastric ulcers, which may also have been present.

The severity of the IBD based on duodenal biopsies seemed to predict the likelihood of improvement after six weeks, since about three-quarters of the horses with mild or moderate IBD improved and only about half of the horses with severe IBD improved. However, this association was not statistically significant, which could be the result of a lack of statistical power because of the small number of severe IBD cases. Of the horses that had improved clinically after six weeks, approximately three-quarters were still alive after one year compared to only 23% of the horses that had not improved after six weeks. In that latter category, nearly 70% of the horses had died or been euthanized for reasons related to IBD. Clinical improvement after six weeks seemed to be a reasonable predictor of one-year survival. Kaikkonen et al. also concluded that the initial response to treatment, with anthelmintics and corticosteroids over a period of three weeks, was a good prognostic indicator for three-year survival in horses with IBD [19].

A second duodenal biopsy was obtained in about 20% of cases. The percentage of horses showing clinical improvement was approximately the same in cases in which the histological severity had decreased as in cases in which this had remained the same or increased. The lack of a correlation between histological and clinical improvements could mean that a duodenal biopsy sample is not representative for the severity or extent of inflammatory changes in the intestine, or that the histological changes are not (directly) related to clinical signs. Another possible explanation could be that histological improvement lags behind clinical improvement and takes more than six weeks to become evident. Based on this study, a second biopsy, taken six weeks after the first, cannot be recommended for predicting the long-term prognosis.

## 5. Conclusions

LPE is a much more common form of IBD in horses than previously thought, and the infiltrating cell types can be identified in duodenal biopsies. Most horses with IBD demonstrate (partial) malabsorption in an oral glucose absorption test, and horses with histologically severe IBD have lower serum protein concentrations than horses with mild IBD. Approximately 70% of horses with IBD had improved clinically six weeks after their diagnosis, and those horses were more likely to be alive after one year than those that had not improved after six weeks. The histological grading of a second duodenal biopsy, six weeks after the first, does not help predict the long-term prognosis.

## Figures and Tables

**Figure 1 animals-14-01638-f001:**
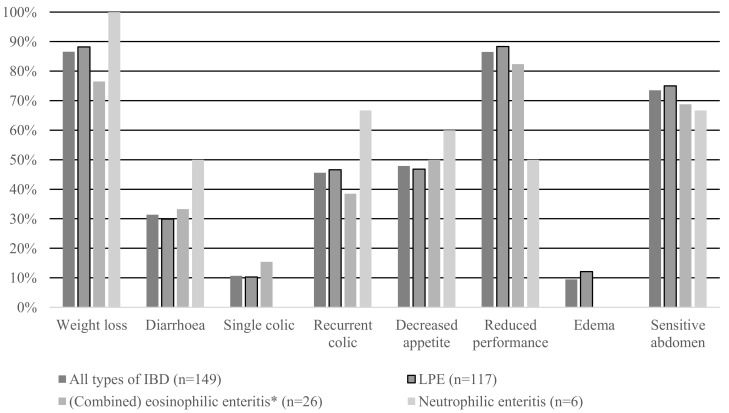
The percentages of horses demonstrating various clinical symptoms for each type of IBD. * The group ‘(Combined) eosinophilic enteritis’ consists of horses with EC, MEED and a combination of LPE and EC.

**Figure 2 animals-14-01638-f002:**
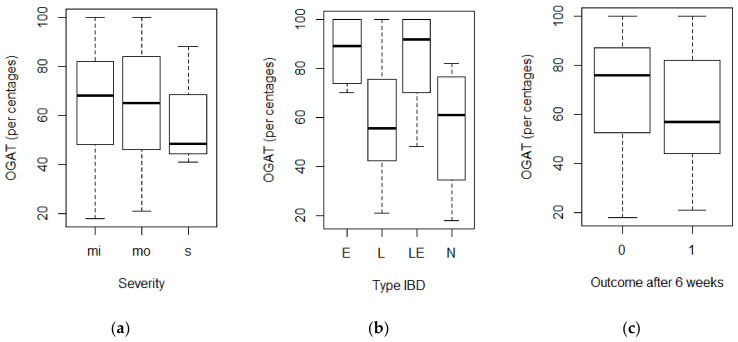
Boxplots of the increases in glucose in percentages during an OGAT in groups of horses with different severity levels (**a**) and types of IBD (**b**) and outcomes after six weeks (**c**). Note: mi = mild; mo = moderate; s = severe; E = EC; L = LPE; LE = LPE and EC; N = neutrophilic enteritis; 0 = not improved; 1 = improved.

**Figure 3 animals-14-01638-f003:**
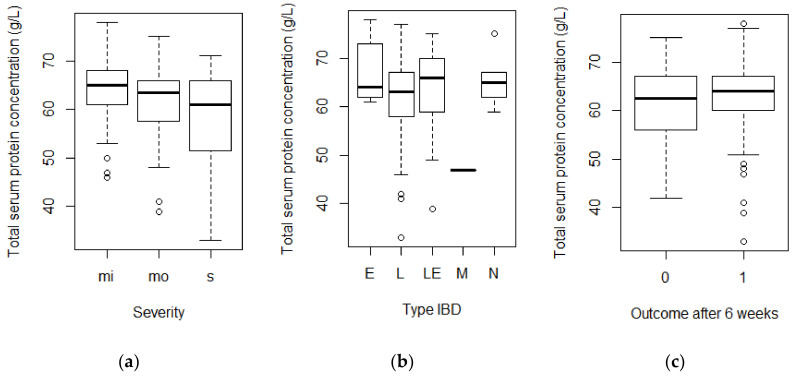
The Boxplots of the total serum protein concentrations in g/L in groups of horses with different severity (**a**) and types of IBD (**b**) and outcome after six weeks (**c**). Note: mi = mild; mo = moderate; s = severe; E = EC; L = LPE; LE = LPE and EC; M = MEED; N = neutrophilic enteritis; 0 = not improved; 1 = improved.

**Table 1 animals-14-01638-t001:** The numbers and proportions of horses with a mild, moderate or severe form of IBD in the different types of IBD.

	LPE (n = 117)	LPE and EC (n = 18)	EC (n = 7)	MEED (n = 1)	Neutrophilic Enteritis (n = 6)
Mild	55 (47.0%)	3 (16.7%)	5 (71.4%)	1 (100%)	6 (100%)
Moderate	54 (46.2%)	10 (55.6%)	2 (28.6%)	0 (0%)	0 (0%)
Severe	8 (6.8%)	5 (27.8%)	0 (0%)	0 (0%)	0 (0%)

**Table 2 animals-14-01638-t002:** The numbers and proportions of horses with ESGD or EGGD in the different types of IBD.

	LPE	LPE and EC	EC	MEED	Neutrophilic Enteritis
ESGD	59 (50.4%)	10 (55.5%)	4 (57.1%)	1 (100%)	4 (66.7%)
EGGD	70 (61.4%)	13 (72.2%)	3 (42.8%)	0 (0%)	3 (50%)

**Table 3 animals-14-01638-t003:** The numbers and proportions of horses with ESGD or EGGD in the different severity groups.

	Mild	Moderate	Severe
ESGD	36 (51.4%)	38 (57.6%)	4 (30.8%)
EGGD	36 (53.7%)	46 (69.7%)	7 (53.8%)

## Data Availability

The raw data supporting the conclusions of this article will be made available by the authors on request.

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
