# Peer review of "Findings and Prognosis in 149 Horses with Histological Changes Compatible with Inflammatory Bowel Disease"

_animals, 2024, doi:10.3390/ani14111638_

Round 1

Reviewer 1 Report

Comments and Suggestions for Authors

This paper looked at clinical and clinicopathological findings in horses with duodenal histological changes indicative of IBD, assessing clinical progression and survival and potential correlation to EGUS.

The study brings a simple and useful contribution to the field of IBD. While in humans and companion animals  this subject is subject to more in depth investigation, in equids there is less information and the lack of insight into its pathophysiology and ethiology limits diagnosis options. Even a simple enumeration of characteristics such as in these case might be helpful for the reader. However, I feel  that a main hypothesis is presented. Do the authors feel like there should be a correlation between all the analyzed variables? The statistics and study design seem a bit chaotic and the paper would benefit from a well written hypothesis. The overall impression is that parameters were just thrown into excel and variant analysis was performed.  The methodology is therefore difficult to understand and a bit vague. Inclusion/ exclusion criteria are not detailed (there is one mention in the results part, that is not its place).

Descriptive statistics are short and while in the statistics description it is mentioned that clinical signs, age, total serum protein concentration, percentage increase of glucose in an OGAT, histological severity, presence of ESGD, presence of EGGD, the type of IBD and the medical treatment on clinical improvement at six weeks was investigated by logistic regression, not all of these are expanded in the results.

There are several limitations to this study. The type of breed representation and the fact that this is a referral clinic. No clear cutoffs for hypoproteinemia, OGAT, lack of all types of IBD, inhomogenous numbers of patients that received treatment so difficult to compare efficiency etc. These are all somewhat discussed but not clearly stated as limitations. However the discussion is interesting and comprehensive and I think this paper is opening some interesting possibilities through its results. 

While this study is retrospective and relies on already sampled biological fluids and diagnostic procedures, please include the fact that (or if) owner consent was requested for publication.

Author Response

Reviewer 1

The study brings a simple and useful contribution to the field of IBD. While in humans and companion animals  this subject is subject to more in depth investigation, in equids there is less information and the lack of insight into its pathophysiology and ethiology limits diagnosis options. Even a simple enumeration of characteristics such as in these case might be helpful for the reader. However, I feel  that a main hypothesis is presented. Do the authors feel like there should be a correlation between all the analyzed variables? The statistics and study design seem a bit chaotic and the paper would benefit from a well written hypothesis. The overall impression is that parameters were just thrown into excel and variant analysis was performed.  The methodology is therefore difficult to understand and a bit vague. Inclusion/ exclusion criteria are not detailed (there is one mention in the results part, that is not its place).

Dear reviewer,

Thank you very much for your valuable suggestions to improve our manuscript.

The introduction has been adapted and we have included a hypothesis.

We have added the inclusion and exclusion criteria in the methods section and removed them from the results.

Descriptive statistics are short and while in the statistics description it is mentioned that clinical signs, age, total serum protein concentration, percentage increase of glucose in an OGAT, histological severity, presence of ESGD, presence of EGGD, the type of IBD and the medical treatment on clinical improvement at six weeks was investigated by logistic regression, not all of these are expanded in the results.

We believe we have covered all these parameters in the results section. Under “Outcome after six weeks” the statistical association between the clinical signs, age, total serum protein concentration, percentage increase of glucose in an OGAT, histological severity, presence of ESGD, presence of EGGD, the type of IBD and the medical treatment and prognosis/survival are presented.

One of the other reviewers indicated tables and figures were missing. These were provided separately (but will be added to the revised version of our paper) and may clarify our results.

There are several limitations to this study. The type of breed representation and the fact that this is a referral clinic. No clear cutoffs for hypoproteinemia, OGAT, lack of all types of IBD, inhomogenous numbers of patients that received treatment so difficult to compare efficiency etc. These are all somewhat discussed but not clearly stated as limitations. However the discussion is interesting and comprehensive and I think this paper is opening some interesting possibilities through its results. 

We have (hopefully) discussed the limitations of our study more clearly/explicitly in the revised paper, including the breeds presented and the fact that our is a referral hospital. Different cut-offs for OGAT are described in the literature and we have discussed the use of various cut-offs for this parameter. 

While this study is retrospective and relies on already sampled biological fluids and diagnostic procedures, please include the fact that (or if) owner consent was requested for publication.

An owner consent form was signed by all owners on admission of their horses to the university clinic, and this has been added to the M&M section. An example of this owner consent form has been sent to the editorial office.

Reviewer 2 Report

Comments and Suggestions for Authors

Dear authors, 

 the work presented in the manuscript is a summary of findings collected from horses diagnosed with IBD in Equine Hospital. The research consisting all information on patients and treatments. In my opinion, the manuscript should be published in a more specialised Journal in the field of Veterinary Science. 

Minor comments:

The simple summary should be rewritten for better flow in a more scientific manner. 

Does the Oral GLucose test perform just due to IBD, were you exclude EMS or other metabolic issues?

Author Response

Reviewer 2

the work presented in the manuscript is a summary of findings collected from horses diagnosed with IBD in Equine Hospital. The research consisting all information on patients and treatments. In my opinion, the manuscript should be published in a more specialised Journal in the field of Veterinary Science. 

The paper has been submitted to a special issue on equine gastro-intestinal health, which we feel is appropriate for this study. Hopefully, the editors will agree.

Minor comments:

The simple summary should be rewritten for better flow in a more scientific manner. 

The simple summary has been slightly rewritten. However, it is intended for a lay audience (as specified by the journal) and we feel it is now appropriate for that audience.

 Does the Oral GLucose test perform just due to IBD, were you exclude EMS or other metabolic issues?

Indeed, the Oral Glucose Absorption Test was performed to investigate small intestinal absorption in the patients, which can be reduced in case of IBD. It is true that this test may be influenced by co-existing metabolic issues such as EMS, but we do not believe that was the case in our study. We have added some discussion about this and indicated that a xylose absorption test may be better/more specific, but also that is not available in our clinic/laboratory.

Reviewer 3 Report

Comments and Suggestions for Authors

Reviewer comments for manuscript ID animals -2966361 entitled ‘Findings and prognosis in horses with histological changes compatible with inflammatory bowel disease’

General comments

Inflammatory Bowel disease is still a poorly understood disease syndrome in equines with multifactorial etiopathologies.  There are many cases of equine colic that are unresponsive to routine therapeutic regimes where IBD seems the underlying cause.  Identifying the underlying cause of colic to IBD and further classifying into different pathological states is indeed very difficult but satisfying for veterinary Pathologists.

The present study is a remarkable work by the authors on the identification of exact cause of IBD in equines. The number of animals studied is sufficient to draw conclusions despite being a retrospective study. The manuscript is well written except  in the discussion section. The authors have described the aims of the study without explicitly reporting the gaps in the studies on IBD. I would like the authors to clearly reports the gaps in literature on IBD before spelling out the aims of the present study for the benefit of the readers.

Statistical methods used in the study are accurate as well as appropriate for this kind of study.

Figures and tables are missing though they have been referred in the text.

The discussion needs improvement as indepth analysis of the results of the present study is needed in light of the previous studies. There are few assumptions made that I have specifically pointed out, without any basis which should have been avoided.

I would like to see the changes in the manuscript as suggested before I can recommend the publication of the manuscript.

Specific comments

Line 64: Please reframe ‘and approximately 90% is less than five years old’ as ‘ with almost 90% aged less than 5 years ‘

Lines 74-76: Please clarify the sentence and rewrite. It is ambiguous.

Lines 87-89: I am sorry I am not able to distinguish the two. Please clarify and rewrite.

Line 93: Please replace ‘sometimes’ with ‘occasionally’

Lines 208-09: In which cases the second/ repeat biopsies were taken. Please clarify.

Line 256: Please delete ‘were’

Lines 272-77: There are contradictory statements in this sentence. Please clarify and please break the sentence into smaller sentences for better comprehensibility.

Lines 305-07: Please clarify whether there was no report of granulomatous enteritis in the study or the cases were not included, if so why !

Lines 360-61: Were these horses subjected to surgery or some other treatment or not treated at all? Please clarify.

Line 456: Please refer to your statement ‘The same is likely to apply to horses’. Why?

Lines 471-72: Please refer to ‘It seems, therefore, that neutrophilic infiltration in cats, dogs and horses is associated with bacteria’ Why ?

Comments on the Quality of English Language

Overall the quality of english is good. However, in some places the sentences are too long that have lead to the creation of ambiguities. 

Author Response

Reviewer 3

General comments

Inflammatory Bowel disease is still a poorly understood disease syndrome in equines with multifactorial etiopathologies.  There are many cases of equine colic that are unresponsive to routine therapeutic regimes where IBD seems the underlying cause.  Identifying the underlying cause of colic to IBD and further classifying into different pathological states is indeed very difficult but satisfying for veterinary Pathologists.

Dear reviewer,

Thank you very much for your valuable suggestions to improve our manuscript.

The present study is a remarkable work by the authors on the identification of exact cause of IBD in equines. The number of animals studied is sufficient to draw conclusions despite being a retrospective study. The manuscript is well written except  in the discussion section.

The authors have described the aims of the study without explicitly reporting the gaps in the studies on IBD. I would like the authors to clearly reports the gaps in literature on IBD before spelling out the aims of the present study for the benefit of the readers.

The introduction has been rewritten to highlight the gaps in the literature and a hypothesis has been added. We hope this makes it clearer why we performed the current study.

Statistical methods used in the study are accurate as well as appropriate for this kind of study.

Figures and tables are missing though they have been referred in the text.

The figures and tables were provided (separately) but will be added to the revised version of the manuscript. This should improve clarity.

The discussion needs improvement as indepth analysis of the results of the present study is needed in light of the previous studies. There are few assumptions made that I have specifically pointed out, without any basis which should have been avoided.

The discussion has been rewritten and specific comments addressed (see also below).

I would like to see the changes in the manuscript as suggested before I can recommend the publication of the manuscript.

Specific comments

Line 64: Please reframe ‘and approximately 90% is less than five years old’ as ‘ with almost 90% aged less than 5 years ‘

Has been amended

Lines 74-76: Please clarify the sentence and rewrite. It is ambiguous.

This has been rewritten to (hopefully) improve clarity

Lines 87-89: I am sorry I am not able to distinguish the two. Please clarify and rewrite.

Has been rewritten

Line 93: Please replace ‘sometimes’ with ‘occasionally’

Has been changed

Lines 208-09: In which cases the second/ repeat biopsies were taken. Please clarify.

This was decided by/together with the owner and this has been added to the paper.

Line 256: Please delete ‘were’

Sentence has been rewritten and comment no longer applies

Lines 272-77: There are contradictory statements in this sentence. Please clarify and please break the sentence into smaller sentences for better comprehensibility.

Different cut-offs for the OGAT are reported in the literature (and this fact has been included in the results section). We have interpreted our findings using the most commonly reported cut-offs. This point is further addressed in the discussion. We hope this is sufficient.

Lines 305-07: Please clarify whether there was no report of granulomatous enteritis in the study or the cases were not included, if so why !

No cases of GE were identified (in any of the examined biopsies) - included has been replaced by identified

Lines 360-61: Were these horses subjected to surgery or some other treatment or not treated at all? Please clarify.

Rewritten and clarified. These horses received no treatment as their symptoms were mild and did not warrant corticosteroid treatment and/or owners were concerned about risk of laminitis with steroid treatment. Has been added (to results section)

Line 456: Please refer to your statement ‘The same is likely to apply to horses’. Why?

As the body’s response to various ‘triggers’ is often similar across species we consider it likely/possible that the same will apply to the horse. We hope this explanation is acceptable.

Lines 471-72: Please refer to ‘It seems, therefore, that neutrophilic infiltration in cats, dogs and horses is associated with bacteria’ Why ?

As neutrophilic enteritis in cats and DPJ in horses are both associated with bacteria it seems reasonable to assume that this may also apply to neutrophilic enteritis in horses. This has been rewritten. Hopefully this is acceptable.

 Comments on the Quality of English Language

Overall the quality of english is good. However, in some places the sentences are too long that have lead to the creation of ambiguities. 

The discussion has been rewritten

The discussion needs improvement as indepth analysis of the results of the present study is needed in light of the previous studies. There are few assumptions made that I have specifically pointed out, without any basis which should have been avoided.

Discussion has been revised

Round 2

Reviewer 1 Report

Comments and Suggestions for Authors

Dear authors, thank you for addressing reviewer comments and improving your manuscript.